# IL-1 Blockade Mitigates Autism and Cerebral Palsy Traits in Offspring In-Utero Exposed to Group B *Streptococcus* Chorioamnionitis

**DOI:** 10.3390/ijms252111393

**Published:** 2024-10-23

**Authors:** Taghreed A. Ayash, Marie-Julie Allard, Mathilde Chevin, Guillaume Sébire

**Affiliations:** 1Department of Molecular Biology and Genetics, Ibnsina National College for Medical Studies, Jeddah 22421, Saudi Arabia; taghreed.ayash@mail.mcgill.ca; 2Department of Pediatrics, Research Institute of the McGill University Health Centre, McGill University, 1001 Decarie Boulevard, Montreal, QC H4A 3J1, Canada; marie-julie.allard@mail.mcgill.ca (M.-J.A.); mathilde.chevin2@mail.mcgill.ca (M.C.)

**Keywords:** GBS, interleukin-1, IL-1Ra, chorioamnionitis, autism spectrum disorder, cerebral palsy, neurodevelopmental disorders, infection, inflammation, brain injury

## Abstract

Group B *Streptococcus* (GBS) is one of the most common bacteria responsible for placental and neonatal infection and inflammation resulting in lifelong neurobehavioral impairments. In particular, GBS-induced chorioamnionitis is known in preclinical models to upregulate inflammatory pathways, primarily through the activation of the interleukin-1 (IL-1) pathway, leading to brain injury and subsequent neurodevelopmental issues. Previous studies from our laboratory using Lewis rat pups have shown that male offspring exposed in utero to GBS chorioamnionitis develop brain injuries leading to neurobehavioral impairments such as autistic traits. In the present study, we aimed to explore whether blocking the IL-1 pathway could prevent or mitigate these neurodevelopmental impairments in adulthood. Using our established preclinical model, we administered IL-1 receptor antagonist (IL-1Ra) to dams with GBS-induced chorioamnionitis. Here, we show that IL-1Ra administration to dams reversed autistic and cerebral palsy traits in male adult offspring exposed in utero to GBS. Hence, IL-1 blockade could serve as a therapeutic intervention against pathogen-induced neurodevelopmental disorders. This research supports the need for future human randomized controlled trials to assess IL-1 blockade administered during pregnancy or in newborns as a strategy to reduce the long-term neurobehavioral consequences of prenatal infections, such as autism, cerebral palsy, learning disabilities, and other neurodevelopmental disorders.

## 1. Introduction

Placental infection or inflammation, most commonly referred to as chorioamnionitis, is a significant medical condition defined by the infiltration of polymorphonuclear cells (PMNs) into the placenta [1]. This condition affects approximately 10% of term infants and up to 60% of very preterm newborns [1,2,3]. Chorioamnionitis often results from a bacterial infection ascending from the lower genital tract, leading to the disruption of the placental barrier and the immune response being triggered [1].

This condition has been shown to have serious implications for neonatal health, particularly when it occurs in preterm infants whose biological systems are less developed [4]. Inflammation and infection can lead to a range of complications, one of the most concerning being perinatal brain injury [3,5,6,7]. Brain injury during the perinatal period—whether from infection, inflammation, or other stressors—can significantly disrupt normal brain development [5,7,8].

Such disruptions in brain development are linked to a range of neurodevelopmental disorders (NDDs), which encompass conditions that affect cognitive, motor, social, and emotional functioning throughout life. These conditions include autism spectrum disorders (ASD), cerebral palsy (CP), learning disabilities, and attention deficit hyperactivity disorder (ADHD), among others [4,5,9,10,11]. The impact of chorioamnionitis on neurodevelopment results in a wide variety of manifestations depending on the severity and timing of the brain injury, with some children experiencing mild developmental delays and others facing lifelong challenges [3,7,12]. These conditions frequently appear in clusters of co-occurring morbidities within individuals, suggesting that NDDs share underlying etiological and pathogenic mechanisms [11,13].

Geneticists have identified numerous genes linked to NDDs, but no single gene has been found to be solely responsible for the most common NDD phenotypes [14,15]. For example, genes associated with specific NDDs contribute to the development of rare syndromes, many of which share overlapping features with other conditions. These genetic factors account for approximately 40% of ASD [15,16,17] and around 10% of CP cases [18]. Similarly, the male-biased sex ratio seen in NDDs, which is only partially explained by X-linked genetic anomalies, is also influenced by sex-specific environmental factors interacting with the genetic background [19,20]. As a result, despite the heritability of some NDDs, it is crucial to explore environmental factors that may interact with or contribute to their onset.

The primary causes of chorioamnionitis are infections due to pathogens such as Group B *Streptococcus* (GBS) and *Escherichia coli* (*E. coli*) [1,3,21]. Epidemiological research has established an association between chorioamnionitis and NDDs [22,23,24]. One key factor in this link is the placental upregulation of the pro-inflammatory cytokine interleukin-1 (IL-1), which has been identified as a significant risk factor for NDDs [23]. Preclinical studies have further elucidated the causal relationship between infections like GBS and the endotoxins (lipopolysaccharides, or LPSs) produced by *E. coli* in inducing chorioamnionitis, which in turn leads to perinatal brain injury. This process is thought to involve the placental production of IL-1, which can disrupt normal brain development and contribute to long-term neurodevelopmental challenges [25,26,27,28].

Beyond the immediate fetal effects, these findings highlight the broad implications of maternal infections and inflammatory responses during pregnancy. Infections like GBS and *E. coli* not only pose risks in preterm birth and neonatal sepsis but also set the stage for complex interactions between immune response and brain development. These insights suggest the need for a deeper understanding of how prenatal inflammation and immune dysregulation can influence long-term neurological outcomes, potentially guiding the development of preventive strategies and interventions for at-risk populations.

Using our Lewis rat model of GBS-induced chorioamnionitis, we observed higher IL-1ß release and PMN recruitment within male placenta than within female placenta [29,30]. Such sex-dimorphic chorioamnionitis is involved in the sex ratio bias towards males in NDDs [28]; male rats exposed in utero to GBS or LPS show poor social communication, sensory modulation, and motor disabilities, displaying key traits similar to those seen in ASD and/or CP patients [25,27]. Female offspring present hyperactive behaviors, mimicking ADHD- and CP-like phenotypes [26,31].

Human recombinant interleukin-1 receptor antagonist (IL-1Ra) is an FDA-approved therapeutic option to treat rheumatoid arthritis and auto-inflammatory disorders [32]. IL-1Ra administration has been shown to be well tolerated during pregnancy [33]. IL-1 blockade using IL-1Ra is currently tested in a phase I/II pilot trial to prevent human preterm lung and brain injuries [34].

Preclinical studies from our laboratory and others already demonstrate the placento- and neuro-protective effects of IL-1 blockade in LPS-induced chorioamnionitis, such as the reduction of neurodevelopmental anomalies [35,36]. We recently investigated whether this maternal treatment could also be suitable in an active bacterial infection, using GBS-induced chorioamnionitis [37]. IL-1Ra administration in Lewis dams provided placento-protective effects without exacerbating GBS infection and prevented short-term neurobehavioral impairments in the offspring, such as lack of maternal attachment [37].

To further investigate the vertical impact of IL-1Ra treatment of GBS chorioamnionitis, we preclinically tested its efficacy for adult male ASD and CP traits.

## 2. Results

### 2.1. Impact of the IL-1Ra Treatment on Maternal and Pup Weight Gain

Dams infected with GBS showed a significant decrease in weight gain (Figure 1a). IL-1Ra administration showed a trend of improvement in weight gain in GBS-exposed dams compared to those exposed solely to GBS (Figure 1a). Pups exposed in utero solely to GBS or to GBS + IL-1Ra presented a postnatal growth impairment (Figure 1b). There was no effect of GBS or IL-1Ra treatment on litter size compared to the control group (Figure 1c). Altogether, IL-1Ra treatment was well tolerated in dams, as well as in pups.

### 2.2. Effects of IL-1 Blockade on Motor and Social Behaviors in Adult Offspring Exposed in Utero to GBS

#### 2.2.1. Effect of IL-1 Blockade on Social Behavior

Social impairments were observed at postnatal day (P)40 in GBS-exposed males in terms of both latency before and number of social interactions. IL-1Ra treatment reverted these GBS-induced social anomalies (Figure 2a,b).

#### 2.2.2. Effect of IL-1 Blockade on Motor Behavior

Motor impairment was observed at P80 in GBS-exposed males in terms of the distance travelled and number of open-field lines crossed as compared to the control group. IL-1Ra administration alleviated these CP traits (Figure 2c,d).

#### 2.2.3. Correlation between IL-1ß Titers at P1 in Pups Exposed in Utero to GBS and Motor Behavior

A significant correlation (*p* = 0.03, R^2^ = 0.36) was observed between IL-1ß blood titers of newborn (P1) males exposed in utero to GBS chorioamnionitis and their mobility at P80 (Figure 2e). This further supports the role of IL-1 as a key determinant in the induction of neuromotor disabilities in male offspring exposed in utero to maternal immune activation.

## 3. Discussion

Our preclinical model, as well as other models of GBS-induced chorioamnionitis, fulfilled criteria of face validity in regard to human ASD and CP traits. In keeping with some of our results, Andrade et al. showed in mice that in utero exposure to GBS in male offspring induced an impairment of mobility at P90 in the open field [38]. The neonatal growth retardation we observed in male rats exposed in utero to GBS is one of the major risk factors for developing ASD and CP manifestations [31]. In the same line, male sex is well recognized in humans as a risk factor for developing CP and ASD [28]. Such sex effects were also observed in the present results. ASD and CP are chronic diseases in humans; interestingly, we show for the first time that in utero exposure to GBS chorioamnionitis leads to chronic ASD and CP traits in our preclinical model.

Our preclinical model of GBS chorioamnionitis has some limitations. It is not a model of ascending chorioamnionitis, which is the most frequent route of human fetal infection. However, such a route of infection induced important intra-litter variability [38]. Behavioral assessment was based on open-field and social interaction tests. Further evaluation of motor and cognitive functions might refine the assessment of the behavioral effect of IL-1Ra. On another note, it would be of interest to perform a neuropathological investigation in our model to gather additional features relevant to adult human ASD and CP.

IL-1Ra is an already approved drug to treat chronic inflammatory conditions, including those affecting pregnant mothers and newborns [35]. IL-1Ra is among the various molecules interfering with the IL-1 signalling and is the one that dominates the pharmacological field due to the following properties:(i)Blocking effect on both IL-1α and IL-1ß.(ii)Short 4–6 h half-life (blood levels falling within a few hours of treatment stoppage).(iii)Multiple routes of administration.(iv)Approval for several pediatric inflammatory conditions and being actually tested in phase I/II therapeutic trial in the premature population [34].(v)Excellent safety record (absence of opportunistic infection, reversible increase in liver enzyme, decrease in polymorphonuclear cells (PMNs), and slight increase in infection, all of which are mostly observed in patients on chronic IL-1Ra treatment) after more than 15 years of use in more than 150,000 patients [39]. In addition, the small size of IL-1Ra and its ability to cross the blood–brain barrier [32,40] make it a very promising neuroprotective molecule in the context of NDDs.

We have already shown that IL-1Ra was a safe and efficient drug to partially prevent LPS- [36] and GBS-induced chorioamnionitis [37] and subsequent short-term neurobehavioral impairments. Demonstrating our ability to optimize such anti-inflammatory treatment in adult animals exposed in utero to GBS brings us to the threshold of preclinical data required for proposing human phase II randomized controlled trials using IL-1Ra (repurposing of FDA-approved IL-1Ra [34,35] treatment during pregnancy) as a placento- and neuro-protective drug with the aim of alleviating the burden of pathogen-induced NDDs.

## 4. Materials and Methods

To further explore the long-term effects of IL-1Ra treatment in GBS-induced chorioamnionitis, we conducted preclinical studies to assess its potential in mitigating adult ASD and CP traits. These traits, which persist into adulthood following early-life neurodevelopmental disruptions, were evaluated to determine whether IL-1Ra administration could provide long-lasting neuroprotective benefits and reduce the severity of these neurodevelopmental outcomes.

As previously described [25,29,37], Lewis dams received a single intraperitoneal injection of live GBS serotype Ia (strain #16955 (generous gift of Dr Claire Poyart); 10^8^ colony-forming units per 100 µL) or saline at gestational day (G) 19. Lewis rats are known to have enhanced susceptibility to inflammatory agents, which is the reason we used them. Human recombinant IL-1Ra (10 mg/kg/12 h) or saline was administered intraperitoneally at 36, 48, and 60 h following GBS exposure, as previously described [37]. The same treatment protocol was already shown to be well tolerated in dams and placento-protective in both LPS [36] and GBS models of chorioamnionitis [37]. The total numbers of animals used in our experiments were 29 dams and 84 offspring. Dams were weighed twice per day and regularly observed to detect any sickness behavior. Maternal weight gain was calculated by subtracting the weight measured at G22 minus the weight at G18. To mitigate the litter effect, cytokine and behavioral outcomes were assessed in one to two animals per litter from both sexes. IL-1ß was measured by Quantikine^®^ ELISA (Bio-Techne, Montreal, QC, Canada) on pups’ sera at P1. Behavioral assessments (open-field and social interaction tests) were conducted on offspring between P40 andP80, as previously described [25,27]. Distance of travel (an assessment of mobility) and the number of lines crossed (an assessment of exploratory behavior) were measured in the open-field test. The social interaction test involved introducing two rats of the same sex and experimental condition but of different litters into the experimental setting. The latency before social interaction and the number of social episodes (sniffing, grooming, chasing, or playing together) were assessed in a restrained enclosure.

Each data set was assessed for the normality of distribution and for outliers using Shapiro–Wilk and Grubb’s tests, respectively. Data were analyzed by one-way analysis of variances (ANOVA), and when data were significant, pairwise comparisons were performed using Tukey’s HSD. Two-way ANOVA was performed to analyze the mean weight of pups between treatments (Control, GBS + Saline, GBS + IL-1Ra) at different times (repeated measures; P1 to P9). Bonferroni’s multiple comparison test was applied if there was a significant interaction between times and treatments at the level of *p* < 0.05. Correlations between IL-1ß concentration at P1 and litter-matched P80 motor behavior (distance in the open field) were analyzed by Spearman correlation, and the goodness of fit was calculated using Graph Pad Prism software version 10 (Graph Pad Software, San Diego, CA, USA). The statistical significance level was set at *p* < 0.05. All figures were constructed using Graph Pad Prism 10.

The experimental protocol was approved by the Institutional Animal Care Committee of the McGill University (protocol #MUHC-7675) in accordance with the Canadian Council on Animal Care guidelines.

## 5. Conclusions

This study is a necessary step of preclinical research before moving to human clinical therapeutic trials focusing either on pregnant mothers with signs of chorioamnionitis or on newborns exposed in utero to chorioamnionitis, for which there is currently no therapeutic intervention that targets the inflammatory burst and its detrimental neurodevelopmental consequences on the offspring.

Here, we provide new evidence of the long-term neuroprotective effects of IL-1 blockade on the behavior of adult animals exposed in utero to GBS infection. Namely, we show improvements in the autistic- and CP-like traits of adult offspring. Combined with previous results describing the placento- and short-term neuro-protective effects of IL-1 blockade, this research paves the way for new avenues of NDD prevention thanks to the adjunction of targeted anti-inflammatory treatment blocking IL-1 to the sole antibiotic treatment currently administered to GBS-infected mothers.

## Figures and Tables

**Figure 1 ijms-25-11393-f001:**
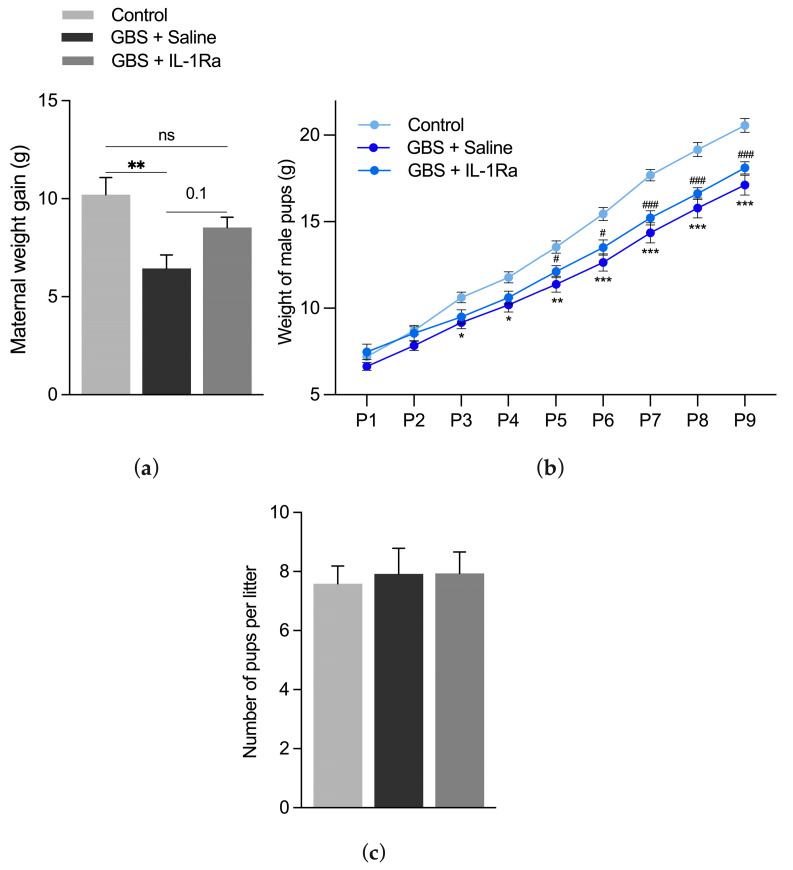
Effect of the IL-1Ra treatment on maternal and pup weight gain and litter size. (**a**) Mean maternal weight gain (g) between gestational day (G)18 and G22 from n = 8–11 dams per experimental group; (**b**) mean male pup weight (g) from postnatal day (P)1 to P9 with n = 8–14 pups per experimental group; (**c**) mean number of pups per litter in 13–17 litters per experimental group. Data were expressed as mean ± SEM, using one-way or two-way ANOVA. ns: non-significant. * *p* < 0.05, ** *p* < 0.01, *** *p* < 0.001 refers to GBS compared to control. # *p* < 0.05, ### *p* < 0.001 refers to GBS + IL-1Ra compared to control.

**Figure 2 ijms-25-11393-f002:**
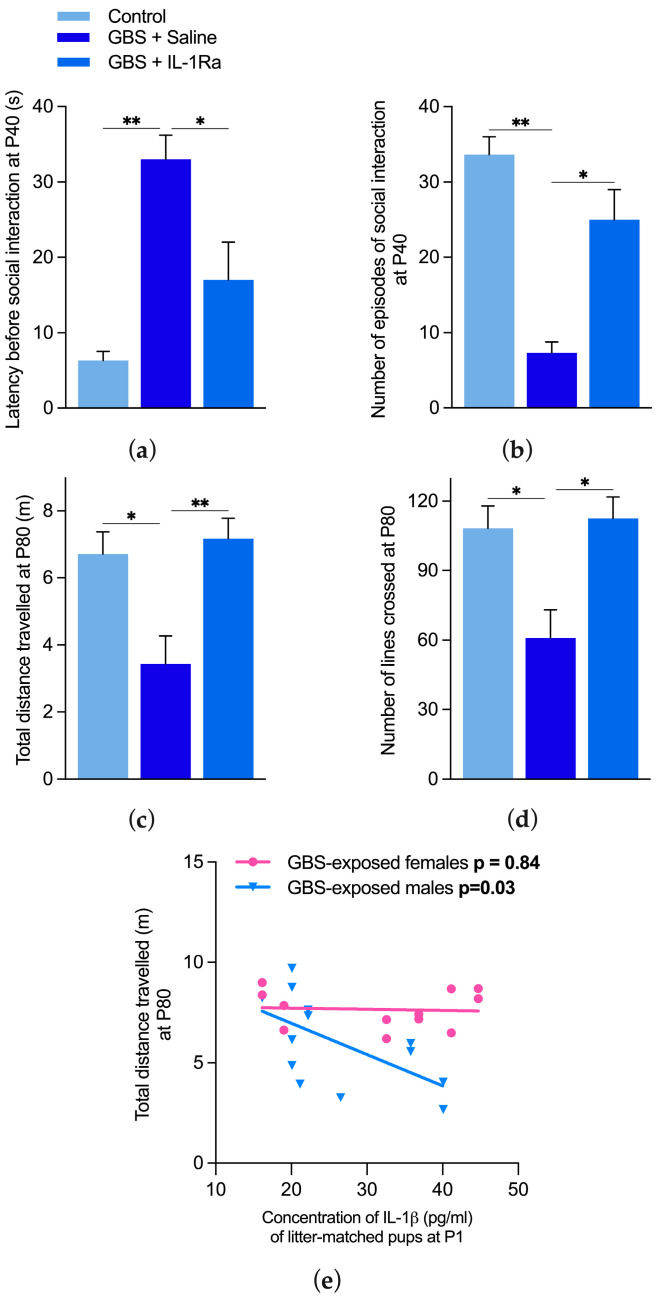
Effect of IL-1Ra on neurobehavioral impairments in offspring exposed in utero to GBS chorioamnionitis. (**a**) The latency before social interaction and (**b**) the number of social episodes were assessed during a 5 min period in an open-field apparatus at P40. (**c**) The travelled distance and (**d**) number of lines crossed were measured during a 5 min period in the open field at P80. Pairs of animals (from 4 to 6 litters) per experimental group were used for the social interaction test. n = 4–9 animals per experimental group from 4–5 litters were used for the open-field test. n = 3–6 animals per experimental group from 4–5 litters were used for IL-1ß titers. Data were expressed as mean ± SEM. * *p* < 0.05, ** *p* < 0.01, using one-way ANOVA. (**e**) A significant correlation was found between IL-1ß blood titers at P1 and the distance travelled at P80 in males but not females, using the Spearman correlation test, *p* < 0.05.

## Data Availability

The data presented in this study are available on request from the corresponding author.

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
