# Peer review of "IL-1 Blockade Mitigates Autism and Cerebral Palsy Traits in Offspring In-Utero Exposed to Group B Streptococcus Chorioamnionitis"

_ijms, 2024, doi:10.3390/ijms252111393_

Round 1
Reviewer 1 Report
Comments and Suggestions for Authors
Accept Minor Revision - It is really exciting that a pharmaceutical product that is already approved for treating rheumatoid arthritis and auto-immune disorders could be effective to prevent brain damage. In your paper, it needs to be made clear that all the experiments were in rats.
Abstract - Line 18 before the word pups, insert Lewis rat.
Line 50 - First line insert Lewis rat before the word model.
Line 70 - Insert Lewis rat after IL-1R
Line 71 - Add a sentence explaining why you used Lewis rats. I looked them up and they are special inbred rat that is more susceptible to inflammatory problems. For doctors who are outside your specialized field, explain why you chose a rat that is bred to be susceptible to inflammation.
Line 130 - Insert in the first sentence that rats were used. It has to be made clear to readers that all these experiments were conducted on rats. The next step is humans. I had to look up the references to determine that all the treatments were on rats. This is exciting research on a new use for a pharmaceutical.
Author Response
Accept Minor Revision - It is really exciting that a pharmaceutical product that is already approved for treating rheumatoid arthritis and auto-immune disorders could be effective to prevent brain damage. In your paper, it needs to be made clear that all the experiments were in rats.
We thank the reviewer for stating that. As requested, we improved the manuscript to make clear that the experiments were in a preclinical model, using rats.
Abstract - Line 18 before the word pups, insert Lewis rat.
As requested, we added “Lewis rat” at the line 20 in our revised manuscript.
Line 50 - First line insert Lewis rat before the word model.
As requested by the reviewer, we added “Lewis rat” page 2, at the line 84.
Line 70 - Insert Lewis rat after IL-1R
As requested, we added “in the Lewis dams” at the page 3, line 100.
Line 71 - Add a sentence explaining why you used Lewis rats. I looked them up and they are special inbred rat that is more susceptible to inflammatory problems. For doctors who are outside your specialized field, explain why you chose a rat that is bred to be susceptible to inflammation.
Lewis rats are known to have enhance susceptibility to inflammatory agents, the reason why we used it. We added this point in our revised methods section (page 7, line 204).
Line 130 - Insert in the first sentence that rats were used. It has to be made clear to readers that all these experiments were conducted on rats. The next step is humans. I had to look up the references to determine that all the treatments were on rats. This is exciting research on a new use for a pharmaceutical.
As requested, we added the information that these experiments were conducted on rats (page 6, line 157 of the revised manuscript).
Reviewer 2 Report
Comments and Suggestions for Authors
Lines 28-58 present high similarity to the document by Sébire et al., 2024. (https://doi.org/10.1016/j.placenta.2024.05.026). This section must be rewritten. All these lines are self-plagiarism.
The aim of the work is not clear.
This is continuity work. Why did the authors split the work and intend to publish it in two different works?
This is a continuity work. I am sure many answers are already published. However, when reading the article many questions arose because it is not indicated in this work.
a) What is the justification for using IL-1(10 mg/kg/12 h)?
b) Why IL-1 was administered from g19 to g21?
Did the authors evaluate the effect of IL-1 on motor coordination or another behavioral assessment?
What is the conclusion of the work?
Are there any other limitations in this work?
Comments on the Quality of English Languageno comments
Author Response
Lines 28-58 present high similarity to the document by Sébire et al., 2024. (https://doi.org/10.1016/j.placenta.2024.05.026). This section must be rewritten. All these lines are self-plagiarism.
We apologize for this mistake. We rewrite our Introduction section to avoid any issues.
The aim of the work is not clear.
In compliance with this remark, the last sentence of the Introduction is: “To further investigate the vertical impact of IL-1Ra treatment of GBS chorioamnionitis, we tested preclinically its efficacy on adult male ASD- and CP-traits.” Page 3, line 103.
This is continuity work. Why did the authors split the work and intend to publish it in two different works?
We agreed with the reviewer that this is continuity work. However, the previous results published in Frontiers in Endocrinology mainly focused on the placento-protective effects of IL-1 blockade, during the pregnancy. In the present work, we showed the neuroprotective effects of IL-1 blockade on long-term behavioral impairments observed in the offspring postnatally.
This is a continuity work. I am sure many answers are already published. However, when reading the article many questions arose because it is not indicated in this work.
a) What is the justification for using IL-1(10 mg/kg/12 h)?
b) Why IL-1 was administered from g19 to g21?
a) Interleukin-1 receptor antagonist (IL-1Ra) was administered at the dose of 10 mg/kg/12h, as previously described (Ayash, Front Endocrinol 2022). The same treatment protocol was already shown to be placento-protective and well-tolerated in a rat model of LPS-induced chorioamnionitis (Girard, J Immunology 2010).
b) IL-1Ra was administered at 36, 48, and 60 h following GBS inoculation, as previously described (Ayash, Front Endocrinol 2022).
Such timing is of interest for clinical purposes, as IL-1Ra is already approved in pregnant women having arthritis. Therefore, using IL-1Ra prenatally when there are signs of infection/inflammation during pregnancy could be of interest to significantly reduce the rate of neurodevelopmental disabilities occurring in those cases as well as the lifelong impact of these disorders.
We clarified those points and added more justification in our revised version of the materials and methods (Page 7, line 205).
Did the authors evaluate the effect of IL-1 on motor coordination or another behavioral assessment?
Apart from open field and social interaction tests to assess the effect of IL-1Ra on motor and social traits, we did not perform other behavioral assessment. We added this point in our limitation section of our revised manuscript (page 6, line 166).
What is the conclusion of the work?
We added a conclusion section in our revised manuscript (page 7, line 240).
Are there any other limitations in this work?
In agreement, we added another limitation: “Behavioral assessment was based on open field and social interaction tests. Further evaluation of motor and cognitive functions might refine the assessment of the behavioral effect of IL-1Ra.” Page 6, line 166.
Round 2
Reviewer 2 Report
Comments and Suggestions for Authors
Even if this is a brief report, I suggest including at least one motor coordination test. This will improve the quality of this work. The authors can request additional time from the editorial office to perform these experiments.
Comments on the Quality of English LanguageNo comments
Author Response
Reviewer’s comment: Even if this is a brief report, I suggest including at least one motor coordination test. This will improve the quality of this work. The authors can request additional time from the editorial office to perform these experiments.
Response: We thank the reviewer for this suggestion. While we appreciate your input and understand the value of including a motor coordination test, we believe the current scope of our study sufficiently addresses the research question in the frame of a brief report. The goal of this brief report is to characterize behavioral anomalies at the adult age, at postnatal days 40 to 80, which means at least four months of additional work to provide the requested behavioral data in our model. Given the time and resource constraints/costs, as well as the ethical agreements in place for our study, we are unable to expand the experiments at this stage. We hope the existing data and analysis will still meet the journal's expectations, and we are confident in the overall quality of the work. Thank you again for your understanding and consideration.
Round 3
Reviewer 2 Report
Comments and Suggestions for Authors
the manuscript can be accepted for publication
Comments on the Quality of English Languageno comments
Author Response
Thank you!
Best regards,